# Complex trait susceptibilities and population diversity in a sample of 4,145 Russians

Dmitrii Usoltsev [1,2,3,4,5], Nikita Kolosov[1,2,3,4,5], Oxana Rotar[1], Alexander Loboda[1,2,3], Maria Boyarinova[1], Ekaterina Moguchaya[1], Ekaterina Kolesova[1], Anastasia Erina[1], Kristina Tolkunova[1], Valeriia Rezapova[1,2,3], Ivan Molotkov[4,5], Olesya Melnik[1], Olga Freylikhman [1], Nadezhda Paskar[1], Asiiat Alieva[1], Elena Baranova[1], Elena Bazhenova[1], Olga Beliaeva[1], Elena Vasilyeva[1], Sofia Kibkalo[1], Rostislav Skitchenko [1], Alina Babenko[1], Alexey Sergushichev [2], Alena Dushina[6], Ekaterina Lopina[6], Irina Basyrova[6], Roman Libis[6], Dmitrii Duplyakov [7,8], Natalya Cherepanova[8], Kati Donner[9], Paivi Laiho [10], Anna Kostareva[1,2], Alexandra Konradi [1,2], Evgeny Shlyakhto[1], Aarno Palotie [3,9,11], Mark J. Daly [3,9,11] & Mykyta Artomov [1,2,3,4,5,9,11] ✉

The population of Russia consists of more than 150 local ethnicities. The ethnic diversity and geographic origins, which extend from eastern Europe to Asia, make the population uniquely positioned to investigate the shared properties of inherited disease risks between European and Asian ancestries. We present the analysis of genetic and phenotypic data from a cohort of 4,145 individuals collected in three metro areas in western Russia. We show the presence of multiple admixed genetic ancestry clusters spanning from primarily European to Asian and high identity-by-descent sharing with the Finnish population. As a result, there was notable enrichment of Finnish-specific variants in Russia. We illustrate the utility of Russian-descent cohorts for discovery of novel population-specific genetic associations, as well as replication of previously identified associations that were thought to be population-specific in other cohorts. Finally, we provide access to a database of allele frequencies and GWAS results for 464 phenotypes.

Linking inherited DNA variation to the disease risks is one of the main goals in modern predictive medicine. Large-scale projects such as the UK Biobank[1], FinnGen[2] and Biobank Japan[3] have made a substantial contribution to the understanding of human biology and the advancement of personalized medicine. The increasing ethnic diversity of genetic studies resulted in the discovery of population-specific

susceptibility loci that could not be identified in other ancestries[4]. Inclusion of understudied populations into biobank initiatives improves fine-mapping accuracy of previously identified GWAS signals and novel risk gene discovery efforts[5].

Genetic studies in the multinational Russian population, which is geographically located at the crossroads of Europe and Asia, could

[1]Almazov National Medical Research Centre, St Petersburg, Russia. [2]ITMO University, St Petersburg, Russia. [3]Broad Institute, Cambridge, MA, USA. [4]The Institute for Genomic Medicine, Nationwide Children's Hospital, Columbus, OH, USA. [5]Department of Pediatrics, The Ohio State University College of Medicine, Columbus, OH, USA. [6]Orenburg State Medical University, Orenburg, Russia. [7]Samara State Medical University, Samara, Russia. [8]Samara Regional Cardiology Dispensary, Samara, Russia. [9]Institute for Molecular Medicine Finland (FIMM), Helsinki, Finland. [10]Finnish Institute for Health and Welfare (THL), Helsinki, Finland. [11]Analytic and Translational Genetics Unit, Massachusetts General Hospital, Boston, MA, USA. ✉e-mail: mykyta.artomov@nationwidechildrens.org

have a potential power to detect historic origins of population-specific variants. Additionally, genetic data from the Russian population could be a powerful source of replication for population-specific associations found in three largest biobanks, UK biobank[1], FinnGen[2], and Biobank Japan[3].

Historically, polygenic trait genetics has been omitted in Russia, resulting in a lack of GWAS studies based on local cohorts. Russian-descent individuals were involved primarily either as a small part of consortia datasets or as a basis for population genetic studies lacking phenotypic information[6–10].

At the same time, there are more than 150 local ethnicities in Russia that would greatly benefit from the local large genetic variation studies. This becomes especially important in light of the lack of transferability of polygenic risk score models between ancestries[11–13]. The aforementioned variety of ethnicities represents the genetic history of populations between Eastern Europeans, Finns, and Asians. For example, a study of whole genome sequencing data previously showed that populations in the northern regions of Russia, west of the Ural Mountains, not only belong to a Finno-Ugric language group but are also genetically close to Finns, while populations in the central regions of western Russia showed similarities with eastern Europeans[14]. Furthermore, close genetic relation between north-western Russians and Finns was also observed in Balto-Slavic speaking populations comparison[15]. In addition, another study showed that Siberian populations separated from other East Asian populations 8800–11,200 years ago and significantly contributed to the formation of Eastern European populations 4700–8000 years ago[16]. Thus, a gene flow from Asia through the Ural Mountains to eastern Europe was hypothesized. Consistently, a notable genetic relationship between Finns and Mongolian tribes was observed[17]. Additional evidence of the great diversity of the Russian population linking European, Asian, and Native American populations was presented in a country-wide study – Genome Russia Project, yet no phenotypic information was collected at that time resulting in the lack of biomedical applications for this data[18].

For studies reflecting population history, a small sample set (~100 individuals) is usually sufficient; however, personalized medicine, biobank assembly and GWAS require a larger sample size and extensive data collection, which is impossible without epidemiological studies. In 2012−2013 in 12 regions of Russia a national study "Epidemiology of cardiovascular diseases in different regions of the Russian Federation" (ESSE-RF) was launched. Within the framework of this study, a stratified multistage random sample of approximately 22,000 residents with blood biobanking and detailed phenotyping was collected[19].

Here, we present the first results of the analysis of clinical and genetic data from three metro areas that participated in the ESSE-RF study: St. Petersburg, Samara, and Orenburg. Our results reflect the genetic structure of the Western Russian population and phenotypic susceptibilities from 4145 participants across 464 phenotypes.

## Results
### Data collection
A cohort of 4800 residents of three areas in Russia – St. Petersburg ($N = 1600$, $N$ males = 573, $N$ females = 1027), Orenburg ($N = 1600$, $N$ males = 656, $N$ females = 944) and Samara ($N = 1600$, $N$ males = 697, $N$ females = 903) were recruited in 2012−2013 through an ambulatory visit to local hospitals and polyclinics (aged $46 \pm 12$ SD years) (Fig. 1a). Detailed phenotypic data and a blood sample were collected. All participants signed an informed consent. The study protocol was approved by local ethics committees (Almazov National Medical Research Center, St. Petersburg) (Methods, Data collection; Supplementary Note 1).

Each patient was invited for an ambulatory visit for one day to collect phenotypic information (Fig. 1b). In 2018−2019, 289 out of 1600

original patients from St. Petersburg were invited for an additional ambulatory visit as a part of different local studies (familial hypercholesterolemia, metabolically healthy obesity, premature vascular aging). These patients were subjected to a detailed follow-up data collection protocol (Methods, Data collection; Supplementary Note 2).

For all participants, biannual updates on phenotypic and vital status were recorded through direct contact (phone calls) and indirect contacts (mail/e-mail letters, information from local clinical district databases, Supplementary Data 1). Additional independent cohort of 138 samples were recruited in 2017−2018 for participation as controls in a local study of early childhood starvation effects. They were evaluated using a similar clinical short protocol, and no follow-up data collection was performed (Supplementary Note 3).

### Genetic data generation and imputation
A total of 4723 individuals, comprising 4594 population samples (ESSE) and 129 starvation controls were genotyped using FinnGen ThermoFisher Axiom microarray[2]. Genotype imputation was conducted using Haplotype Reference Consortium (HRC) panel[12,20] and Beagle 5.2 software[21] (Methods. Genetic data generation and imputation). The quality of the imputation was assessed using a masking experiment (Supplementary Note 4.1, Supplementary Fig. 1).

Samples with sex-mismatch (224 ESSE and 28 Starvation Controls) (Supplementary Materials, Supplementary Note 4.2, Supplementary Fig. 2) and duplicated samples (190) were excluded. Additionally, 347 related individuals were marked for further analysis (Supplementary Note 4.3, Supplementary Fig. 3).

The final dataset included 4281 individuals (4183 ESSE and 98 Starvation Controls) and 11,077,763 variants. Overall, 37,439 variants failed Hardy–Weinberg equilibrium ($p < 1 \times 10^{-4}$) and 371 variants were discordant with the HRC imputation panel. Allele frequencies were compared against gnomAD Finnish and non-Finnish European populations (Supplementary Note 4.4, Supplementary Fig. 4).

### Population structure analysis
Initially, we performed principal component analysis (PCA) to identify ancestral clusters within the dataset (Methods; Population structure analysis; Fig. 2a; Supplementary Note 5; Supplementary Figs. 5 and 6). Joint PCA with the 1000 Genomes dataset indicated that the western Russian population is represented by individuals of admixed ancestry, spanning from European to East Asian continental ancestry (Fig. 2b, Supplementary Fig. 7a). A separate joint PCA with only European subpopulations from 1000 Genomes demonstrated closer relatedness of Russians to Finnish population. This relationship was particularly evident in the overlap observed at the PC1:PC2 plane (Supplementary Note 6, Fig. 2c, Supplementary Fig. 7b). We were able to distinguish between Russians and Finnish populations at higher principal components (PC3-PC4; Supplementary Fig. 8). Six clusters within the Russian data set were identified in the PCA space (Supplementary Note 7, Fig. 2d, Supplementary Fig. 9).

We calculated the contribution of haplotypes from geographically proximate ancestries represented in 1000 Genomes to each cluster using the ADMIXTURE[22] (Supplementary Note 8, Fig. 2e, Supplementary Fig. 10). The first cluster included mostly individuals related to southern Europeans. The second cluster predominantly included a population close to Utah residents (CEU) with Northern and Western European ancestry and the British with a small admixture of Finnish and Asian ancestry. In the next four clusters, the proportion of Finnish and Asian haplotypes increased and the proportion Northern and Western European ancestry haplotypes decreased, which was consistent with the location of these clusters in the principal components space; these clusters co-localized with the Finnish population in PC1-PC2 (Supplementary Note 6, Fig. 2c). Furthermore, in order to establish the independence of our findings from the genotyping platform, we conducted an ADMIXTURE analysis exclusively utilizing HapMap

**a**

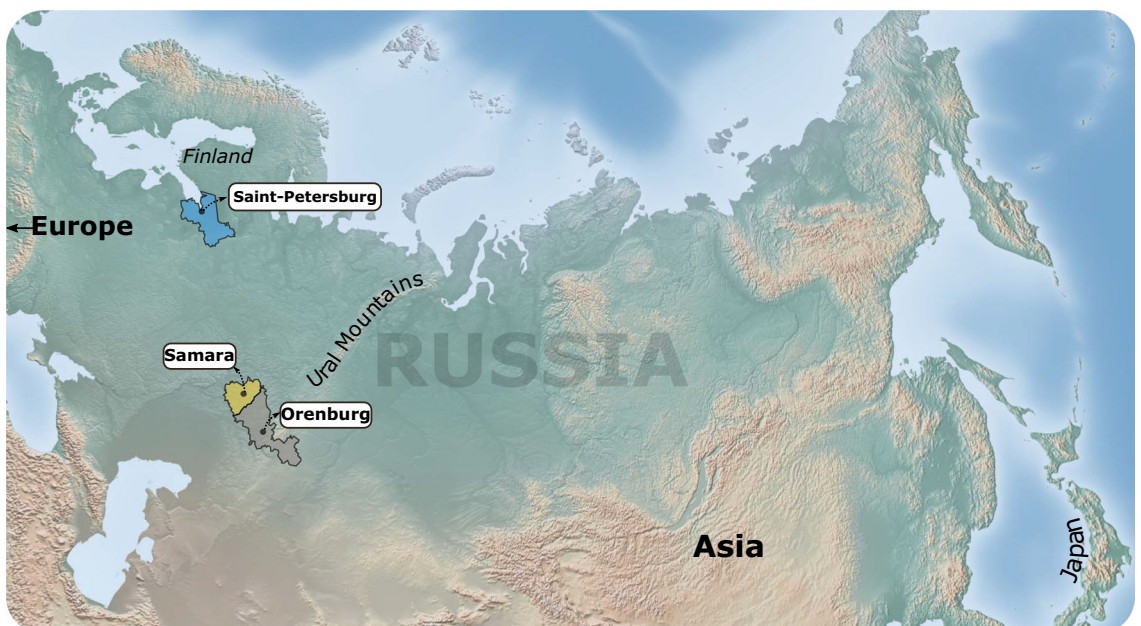

**b**

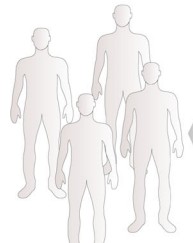

Fig. 1 | Cohort description and study design. a Areas in the European part of Russia where sample collection was conducted. The map was generated with the Matplotlib (© 2011-2014 Jeffrey Whitaker. © 2015–2024 The Matplotlib development team) Basemap toolkit for Python (https://pypi.python.org/pypi/basemap); b Description of the available genetic, clinical, self-reported and follow-up data. Biobank Russia logo was created by Nikita Kolosov.

variants. The outcome of this analysis revealed no qualitative alterations in our observations (Supplementary Fig. 11).

Next, $F_{st}$ and IBD-sharing statistics between Russians and other populations from 1000 Genomes were calculated. We found that according to the $F_{st}$ Russian population in general is close to CEU (Supplementary Fig. 12). However, a more precise cluster comparison showed, for example, that cluster 4 was close to Finnish population and cluster 6 was closest to Asian populations which was consistent with PCA and ADMIXTURE analysis (Supplementary Note 9, Supplementary Fig. 13).

To verify the relationship between Russians and Finns we calculated IBD-sharing statistics between all Russian clusters and 1000

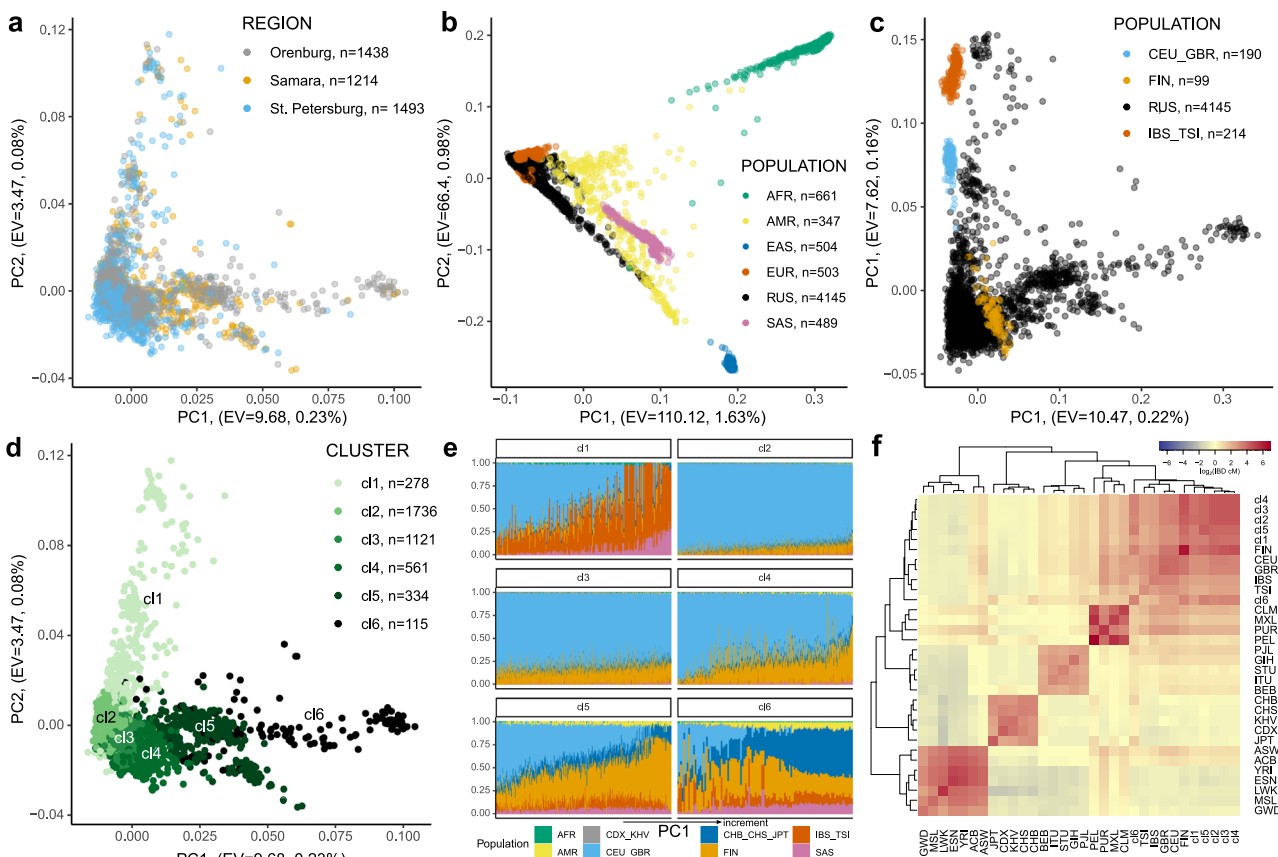

**Fig. 2 | Population structure analysis.** The following abbreviations were used: *AFR* Africans (ACB African Caribbean in Barbados, ASW African Ancestry in SW USA, ESN Esan in Nigeria, GWD Gambian in Western Division – Mandinka, LWK Luhya in Webuye, Kenya; MSL Mende in Sierra Leone; YRI Yoruba in Ibadan, Nigeria), *AMR* native Americans (CLM Colombian in Medellín, Colombia, MXL Mexican Ancestry in Los Angeles CA USA, PEL Peruvian in Lima Peru, PUR Puerto Rican in Puerto Rico), *EAS* east Asians (CHB Han Chinese, CHS Han Chinese South, JPT Japanese in Tokyo, CDX Chinese Dai in Xishuangbanna, KHV Kinh in Vietnam), *RUS* Russians, *SAS* south Asians (BEB Bengali in Bangladesh, GIH Gujarati Indians in Houston, Texas, USA; ITU Indian Telugu in the U.K.; PJL Punjabi in Lahore, Pakistan; STU Sri Lankan Tamil in the UK), *EUR* Europeans (CEU Utah residents (CEPH) with Northern and Western European ancestry, GBR British, FIN Finnish in Finland, IBS Iberian populations in Spain, TSI Toscani in Italy). **a** Principal component analysis with labeling indicating sample collection region; **b** Joint principal component analysis with 1000 Genomes cohort; **c** Joint principal component analysis with European subsample of 1000 Genomes cohort; **d** Clustering of the Russian cohort in the PCA space; **e** Admixture analysis for each of six populational clusters in the Russian cohort, samples are arranged with respect to their PC1 coordinate; **f** Hierarchical clustering of the Russian cohort with 1000 Genomes subpopulations with respect to the sharing of IBD regions.

Genomes populations. We collected all IBD regions with LOD score quality greater than 3 for each pair of individuals. Next, we merged the IBD regions if the gap was not greater than 0.6 cM and if there was not more than 1 discordant homozygous variant. We summarized the length of resulting IBD regions for each pair of individuals and calculated the median IBD length between pairs of populations (Supplementary Fig. 14). The resulting heatmap for all populations is shown in Fig. 2f. According to the clustering analysis, the Finns are closer to the first five Russian clusters than to the European populations from 1000 Genomes (Supplementary Note 10).

Conclusively, the Russian population sampled in the large metro-areas represents a heterogeneous combination of individuals of admixed ancestries between European and Asian populations. In addition, for a subgroup of the sixth cluster with the highest PC1 values more than half of the haplotypes are of Asian origin. This finding is expected, as these samples primarily come from Orenburg, a region located near the border with Kazakhstan (Central Asia). It has previously been shown that individuals from Central Asia have a mixture of European and Asian haplotypes[23–25].

To eliminate the potential influence of systematic errors linked to imputation, we conducted a population structure analysis solely utilizing genotyped variants. Notably, the comparison between cluster assessments based on genotyped variants and assessments involving all variants revealed a robust correlation ($R^2 = 0.87$). This correlation highlighted the strong alignment between these two approaches. As a result, the outcomes of our population structure analysis provided strong validation for the consistency of our observations, underscoring that imputation had no significant effect on the results (Supplementary Note 11, Supplementary Fig. 15).

## Enrichment of finnish and russian variants

The analysis of population structure revealed an intricate pattern of relatedness among Russians, Finns, and East Asians. The Finnish population is historically unique; however, the genetic similarity of the Russian population illustrated above suggests that DNA variants enriched in Finns might also be found in Russia.

To assess the population-specific properties of DNA variants, we created the distributions of $\log_2$ allele frequency ratios between the target population and non-Finnish Europeans from gnomAD (NFE) for 265,624 Finnish enriched variants (Methods. Enrichment of Finnish and Russian variants; Fig. 3a, Supplementary Fig. 16). The medians of these log-ratio allele frequency distributions (excluding variants that were not observed) increased from cluster 2 (0.56) to cluster 6 (3.05). In the sixth cluster, it reached an even higher value than in the Finnish population (3.032). As the fraction of East Asian haplotypes increases from cluster 2 to cluster 6 of the Russian

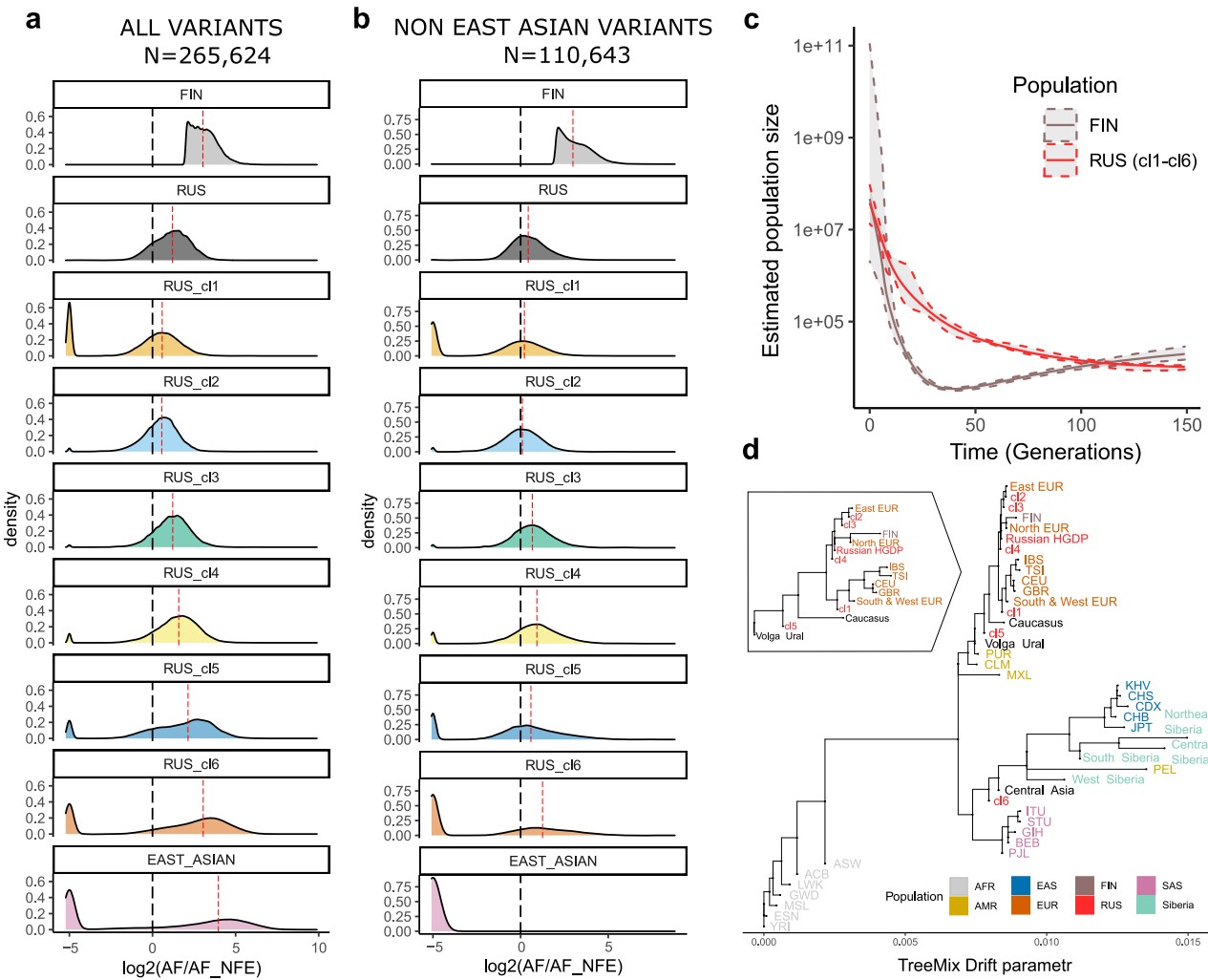

**Fig. 3 | Analysis of prevalence of Finnish-enriched variants in the Russian cohort and TreeMix population structure analysis. a** Distribution of enrichment of the Finnish-enriched variants across clusters in the Russian cohort, Finnish and East Asian cohorts from gnomAD; **b** Distribution of enrichment of the Finnish-enriched variants that are not found in East Asian population in gnomAD across clusters in the Russian cohort and Finnish cohort from gnomAD. Red dashed line shows the median of the distribution; **c** Analysis of estimate population size for Finnish subpopulation from 1000 Genomes cohort and the Russian cohort, error bands show 95% bootstrap confidence interval; **d** TreeMix analysis of the relatedness between 1000 Genomes subpopulations, EGDP populations, HGDP Russians and clusters within the Russian cohort.

population, we investigated the distribution of log-ratios between allele frequencies of Finnish-enriched variants in East Asians compared to Non-Finnish Europeans (Fig. 3a, Supplementary Data 2). The median value of enrichment was 3.962 which was more than observed in the Finnish population. We found that a significant part (58.35%, $N = 154,981$) of the Finnish-enriched variants have a non-zero frequency in the East Asian population. Furthermore, 45.25%, $N = 120,204$ variants have a log-ratio greater than 2, indicating that these variants are, in fact, more frequent in East Asians than in Finns. In order to assess any potential bias linked to imputed variants, we conducted a frequency analysis on directly genotyped variants. The outcomes obtained were consistent with our earlier findings (Supplementary Fig. 17).

To investigate the prevalence of Finnish-enriched variants in the Russian sample, we selected only 110,643 variants that were not observed in the East Asian population in gnomAD and found that the highest median enrichments in the Russian cohort were observed in clusters 3–6, which represent the admixed Finno-Asian haplotype

structure (Supplementary Note 12; Fig. 3b, Supplementary Data 3). Previously reported Finnish-enriched variants associated with clinical phenotypes were generally more prevalent in the Russian population compared to NFE (Supplementary Fig. 18)[2]. The summary of VEP annotations for Finnish-enriched variants is shown in Supplementary Fig. 19a, b.

An increase in haplotype frequency usually occurs as a result of the "bottleneck." For example, for the Finnish population, several bottlenecks were previously reported[26]. We estimated the population size history for the Russian cohort using Finnish samples from the 1000 Genomes as a comparison (Fig. 3c). No bottleneck was observed in the Russian population, despite the presence of Finnish-enriched variants. We also computed population size based on the cluster structure in the Russian population and observed a sign of a bottleneck for clusters with the largest fraction of Asian ancestry (Supplementary Note 13; Supplementary Fig. 20).

We sought to understand the historical origins of the Russian subpopulations and their connection to Finnish and Asian populations,

which would potentially explain the presence of population-specific variants. We calculated a maximum likelihood tree using the TreeMix model with reference populations: 1000 Genomes; the populations closest to Russia from the Estonian Genome Diversity Panel (EGDP)[8]; and Russians from the Human Genome Diversity Panel (HGDP)[7]. Interestingly, Finnish samples were the closest to Russians from HGDP and cluster 4, while cluster 5 was close to Ural populations (Fig. 3d). The only Russian cluster within the Asian branch is cluster 6, which is further supported by its close proximity in principal component space to Central Asia (Supplementary Fig. 21). Furthermore, we performed Treemix analysis additionally using all HGDP populations (Supplementary Note 14; Supplementary Fig. 22).

Using the ADMIXTURE in unsupervised mode with 8 clusters we built clusterization consistent with our tree, and detected a genetic component (yellow) widely represented in the Siberian population. The percentage of this genetic component decreased with moving from Siberia to East Europe through the Urals. Also, this component is present in the Finnish population, which may be the result of a known gene flow from Asia to Europe through the Ural Mountains (Supplementary Fig. 23).

The Russian population also had enrichment of population-specific variants compared to the Europeans, some of which have been described previously[18]. We identified 44,936 Russian-enriched variants as having a log-ratio between RUS and NFE from gnomAD greater than 2. Among Russian-enriched variants 40,743 (90.61%) had higher AF in East Asians than in Russians and only 2045 (4.55%) were not observed in the East Asian population in gnomAD (Supplementary Fig. 19c, d).

## GWAS

We performed GWAS for 464 phenotypes. Results are available at https://biobank.almazovcentre.ru.

Although 4600 samples is a relatively modest cohort size for GWAS, we provide examples of replication of findings from other biobanks as well as newly identified associations specific to the Russian population.

Several associations observed in the UK biobank were replicated, for example, rs7412 and rs4970834 for LDL, rs4697701 and rs4549940 for uric acid levels (Supplementary Fig. 24a, b).

Interestingly, nominal associations in the UK biobank, such as the association of rs13266066 with the initiation of smoking (UKBB phenocode 20116_0, beta = −0.0043, $p = 0.00022$)[1], later confirmed with the MTAG approach using multiple addiction phenotypes (beta = −0.007, $p = 1 \times 10^{-10}$, beta was reversed to match models)[27], were significantly associated with the "never smoked" phenotype in the Russian cohort (N never smoked = 2391; AF never smoked = 0.414, N controls = 1488; AF controls = 0.475, beta = −0.28, $p = 3.74 \times 10^{-8}$, Supplementary Fig. 24c). To reduce the possibility of technical artifacts associated with this observation, we looked at only directly genotyped variants in this locus and confirmed the presence of highly-associated rs11781072 ($p = 1.45 \times 10^{-6}$). eQTL properties of rs13266066 are associated with expression of *PTK2* in cerebellum ($p = 8.8 \times 10^{-12}$). We also performed a gene prioritization analysis which indicated the putative causal role of *PTK2* (Supplementary Fig. 25).

Several novel genome-wide significant associations were identified: current smoking (rs7972723, with AF of 0.1465 in RUS, 0.1447 in NFE, and 0.1680 in FIN populations; 834 individuals as current smokers, AF of 0.189, and 3045 controls with AF of 0.137; beta = 0.43, $p = 2.08 \times 10^{-8}$; Supplementary Fig. 24d), abdominal obesity (rs56046524, with AF of 0.3537 in RUS, 0.3938 in NFE, and 0.3632 in FIN populations; 1405 cases with allele frequency of 0.306, and 2462 controls with AF of 0.378; beta = −0.324, $p = 3.7 \times 10^{-9}$; Supplementary Fig. 24e), and increased blood pressure in the second half of pregnancy (rs11948871, with AF of 0.2024 in RUS, 0.1727 in NFE, and 0.1479 in FIN populations; 366 cases with AF of 0.279, and 1642 controls with AF of 0.185; beta = 0.55, $p = 1.4 \times 10^{-8}$; Supplementary Fig. 24f). However,

given the modest size of the discovery cohort, thorough replication is necessary for these findings (Supplementary Note 15).

Due to the presence of six genetic clusters in our population, we analyzed the effect sizes of the discussed variants in each cluster independently. We observed overall consistency in effect sizes. Some clusters were too small to detect a significant deviation of the effect size from 0 (Supplementary Fig. 26).

Population structure in the Russian cohort indicated that it could be feasible to use it for replication of Finnish-specific genetic associations. Despite cohort size limitations, we attempted to illustrate this through a systematic approach. First, we selected only potentially replicable (MAF RUS > 0.01) Finnish-enriched variants that were less frequently observed in the East Asian population than in Finns (log-ratio between EAS and NFE < 2). There were 20,050 such variants, which were not clumped together (includes LD-correlates). Among them, we found 142 variants with genome-wide significance associations with 177 traits in FinnGen (total 773 variant-phenotype pairs). Overlaps with the Russian cohort were found for 62 variants in 53 traits (332 variant-phenotype pairs, Supplementary Fig. 27a).

We combined the phenotypes by similarity into 8 phenotypic groups: Diabetes, Sleep apnea, Asthma, Statin, Alzheimer, Hypothyroidism, Arthritis, Hypertension, and for each group, we independently performed LD-clumping ($R^2 < 0.1$; Supplementary Data 3). The resulting data set contained 11 independent variants in 8 phenotypic groups. The only variant, rs74800719, passed the Bonferroni-adjusted replication threshold ($p = 0.05/11$ variants/8 traits = $5.7 \times 10^{-4}$). This variant was associated with an increased risk of Alzheimer disease in FinnGen ($p = 1.14 \times 10^{-32}$, beta = 0.5718). In the Russian cohort, it was associated with an increase in the comorbid phenotype - apolipoprotein B levels ($p = 4.4 \times 10^{-4}$, beta = 0.15) (Supplementary Fig. 27b)[28].

Furthermore, we conducted an investigation of the genetic correlations between GWAS from the Russian Biobank and their corresponding counterparts in the UK Biobank and FinnGen datasets. Initially, we excluded all GWAS that had heritability estimates outside the range of [0,1] and a heritability standard error more than 50%. This left us with a total of 35 GWAS, 26 of which were matched with corresponding traits from the UK Biobank (Supplementary Data 4). For complex phenotypes, we compared them with all relevant components from the UK Biobank dataset. Thus, we had 34 pairwise comparisons. Out of 34 comparisons, 26 were nominally significant ($p < 0.05$), and 22 passed the Bonferroni significance threshold (0.05/ 34 = 0.00147). From the FinnGen traits, we specifically chose dyslipidemia, hypertension, type 2 diabetes (T2D), obesity, myocardial infarction, ischaemic heart disease, anxiety, depression, smoking, and sleep apnoea. We proceeded to examine the genetic correlation between these selected traits and their corresponding counterparts from our pool of 35 traits. This led to the construction of 18 pairs, with 15 of them showing nominal significance and 8 passing the Bonferroni significance threshold (0.05/18 = 0.00278) (Supplementary Data 4).

## Discussion

The Russian biobank resource presented here is an essential step towards accessibility of precision medicine for the patients with a wide variety of genetic makeups not currently represented in other major genetic studies.

Our cohort illustrates that the genetic structure of the Russian population, sampled in metropolitan areas in the European part of the country, consists of the number of subpopulations with high relatedness to Finnish and East Asian populations. We also identified a subgroup that has Central Asian origins. This subgroup exhibited the highest proportion of Asian haplotypes and represents a mixed population with significant genetic similarity to Central Asian populations. This finding is supported by the geographic distribution of the samples and previous studies on Central Asian populations[23–25,29–31]. The Finno-Ugric subpopulations in Russia are historically found west

of the Uralic mountains, which is in good agreement with previous whole genome studies conducted in this area[9,14]. The gene flow from Asia through Siberia and the Ural Mountains to eastern Europe, together with the previously found relationship between Finns and Asians, suggests that the unique Finnish variants could also be found on the territory of modern Russia. The bottleneck that Finnish population went through resulted in an increase of the allele frequencies of common Finno-Ugric ancestor population. Moreover, our IBD and ADMIXTURE analyses provide potential explanations to high relatedness between Finnish and Mongolian populations reported previously[17].

Replication studies presented here indicate that even with a relatively modest cohort size, previously reported associations from the UK biobank[1] and FinnGen[2] could be directly observed in the Russian cohort. Such a unique genetic structure of the Russian population provides a potential power to discover and replicate associations that often were considered population-specific. Importantly, such replication can be achieved simultaneously with relatively modest cohort size and resources.

This suggests that the susceptibility to polygenic diseases in Russia could potentially be driven by a mixture of variants from multiple ancestral populations. Some of the ancestral populations in this case have not been studied before. Assessing population-dependent contributions of many associated alleles would be critically important for creating informative polygenic risk score models for individual inherited risk evaluation[32].

We continue to monitor participants of this study with the latest data update in the Fall of 2023. Our proof-of-concept study shows that infrastructure, logistics, and research resources are sufficient to create polygenic trait studies in Russia. The major challenge, yet to be resolved, is an outline of how to scale such efforts to the size of other major biobanks.

Finally, we anticipate that the first local resource for polygenic trait genetics studies in Russia, which provides the largest public reference for allele frequencies and genetic associations – Biobank Russia (https://biobank.almazovcentre.ru) – will become a core for further expansion of complex trait genetics research to yet understudied populations.

## Methods
### Data collection
Overall, 4800 patients were invited for one day ambulatory visits in 2012–2013. Several types of data were collected: responses to the health and medical history questionnaire, dietary behavior, physical activity, social status, depression, anxiety and perceived stress tests, anthropometric parameters, blood pressure and heart rate measurements, blood metabolic panel test, including glucose, creatinine, uric acid, total cholesterol, high-density lipoprotein (HDL), low-density lipoprotein (LDL), triglycerides, insulin and N-terminal prohormone of brain natriuretic peptide (pro-BNP) levels measurements.

For the St. Petersburg cohort, several additional phenotypes were collected: measurements of blood pressure and heart rate in standing position, vessel stiffness measurements and electrocardiogram (ECG), urine albumin and extended blood metabolic panel test with additional test for C-reactive protein (CRP), lipoprotein (a), apolipoproteins A and B, cortisol, leptin, adiponectin, and vitamin D. (Supplementary Note 1, Fig. 1b).

### Genetic data generation and imputation
DNA was extracted from blood samples of 4723 individuals (4594 population sample ESSE + 129 Starvation Controls) using the QIAamp DNA Mini Kit (Qiagen) and genotyped using a custom FinnGen ThermoFisher Axiom microarray[2].

Genotype imputation in the Russian population was carried out with the Haplotype Reference Consortium (HRC)[20] reference panel in accordance with optimal solution for the Russian-descent cohorts[12] (Supplementary Note 4.1). The HRC panel was preprocessed and filtered to meet the data formatting requirements for further imputation procedure. After the filtering there were 37,620,211 variants and 27,165 individuals left in HRC.

Before the imputation, genotypes of the study cohort were filtered, pre-phased, strand-checked, and split into individual chromosomes. In the end, we kept 474,430 variants and 4723 individuals for further SNP imputation.

All imputations were performed by Beagle 5.2[21] with the default parameters (burnin = 6, iterations = 12, imp-segment = 6, ne = 1000000). The imputation quality for each variant was measured using Dosage-R2 (DR2), as given in Beagle output. All variants with DR2 >= 0.8 were considered well-imputed and kept for further analysis. We confirmed the quality of imputation using a masking experiment (Supplementary Note 4.1 and Supplementary Fig. 1). 224 ESSE and 28 Starvation Controls samples with mismatch between detected and reported sex were excluded (Supplementary Note 4.2, Supplementary Fig. 2).

Relatedness and sample duplications were assessed for individuals using kinship analysis with PLINK2[33]. 190 duplicates (187 ESSE and 3 Starvation Controls) were removed, and 347 related individuals were labeled for exclusion in further analysis (Supplementary Note 4.3, Supplementary Fig. 3).

The resulting genetic dataset consisted of 4281 (4183 ESSE and 98 Starvation Controls) individuals and 11,077,763 variants. The genetic data was subjected to quality filtration using python3 "hail" (v0.2.85) package[34]. Overall 37,439 variants failing Hardy–Weinberg equilibrium ($p < 1 \times 10^{-4}$) were eliminated from the analyses. Additionally, we removed 371 variants discordant from the HRC imputation panel. The rest of the variants were subjected to comparison of allele frequency against gnomAD Finnish and non-Finnish European populations (Supplementary Note 4.4, Supplementary Fig. 4).

### Population structure analysis
PCA was performed with PLINK2 using 535,727 common autosomal LD-pruned variants ($R^2 < 0.2$). Additionally, we excluded 136 individuals from the ESSE cohort as PCA outliers using "adamethods" (v1.2.1) R package with built-in max Euclidean distances approach[35,36] (Supplementary Note 5, Fig. 2a, Supplementary Fig. 5-6). Then, we merged the genotyping dataset of ESSE/Starvation Controls with 1000 Genomes WGS data and performed PCA on common 506,617 LD-pruned variants using the python3 "hail" (v0.2.85) package (Supplementary Note 6, Fig. 2b, Supplementary Fig. 7a). Additionally, we selected ESSE/Starvation Controls and only 1000 Genomes European populations and ran PCA on common 515,649 LD-pruned variants (Supplementary Note 6, Fig. 2c, Supplementary Figs. 7b and 8). Eigenvalues and the corresponding percentage of variance explained for all PC components were calculated using PLINK2 with the number of PCs equal to the number of individuals in a genotyping dataset.

PCA clusters of Russian samples were identified for 4145 individuals using "SVDFunctions" (v1.2) R package[37] (Supplementary Note 7, Fig. 2d, Supplementary Fig. 9). Admixture analysis was performed for combined (ESSE/Starvation Controls – 1000 Genomes) LD-pruned genotype matrix excluding 361 relative individuals (347 – ESSE, 14–1000 Genomes). The final dataset for ADMIXTURE (v1.3.0) consisted of 6649 individuals and 506,617 variants[22] (Supplementary Note 8, Supplementary Figs. 10 and 11, and Fig. 2d).

We calculated $F_{st}$ statistics using VCFtools (v0.1.15)[38] (Supplementary Note 9, Supplementary Figs. 12 and 13).

The IBD-sharing statistic was calculated for a combined LD-pruned genotype matrix (ESSE/Starvation Controls – 1000 Genomes) lacking related individuals using BEAGLE 4.0 (beagle.r1399.jar)[39] (Supplementary Note 10, Fig. 1f, Supplementary Fig. 14). Estimated population size was calculated for resulting IBD regions with length

more than 2 cM using IBDne (ibdne23Apr20.ae9.jar) tool[40]. Of note, Finnish population size was evaluated only in 1000 Genomes samples, and the small sample size results in larger confidence intervals in the first 10 generations (Supplementary Note 13, Supplementary Fig. 20). The population maximum likelihood tree, based on allele counts for each population, was constructed using TreeMix (v.1.12)[41]. Additional admixture analysis in unsupervised mode was done using ADMIXTURE (v1.3.0) (Supplementary Note 14). Also we performed all populational analyses using only genotyped variants to eliminate the possibility of imputation artifacts interfering with the results (Supplementary Note 11, Supplementary Fig. 15).

### Enrichment of Finnish and Russian variants
For each variant in the Russian dataset with HWE > 0.0001, we obtained an allele frequency in Non-Finnish Europeans (NFE) and Finnish (FIN) samples from gnomAD. Then, we calculated the log-ratio between allele frequencies in FIN and NFE – the higher it is, the more the corresponding variant is specific to the Finnish population. Only variants passing gnomAD RF filters with call rate >= 0.97 were used in this analysis. We considered 265,624 variants with log-ratio greater than 2 and Finnish AF between 0.01 and 0.1 to be Finnish-enriched (Supplementary Fig. 16a)[2].

For these variants, we calculated the log-ratio between Russian (RUS) AF and gnomAD NFE AF and found that they were also enriched in the Russian population (logistic regression: RUS(1)/NFE(0) ~ $\log_2(AF)$; $p < 1 \times 10^{-16}$, beta = 0.15767, se = 0.00046) (Supplementary Fig. 16b). Since the Russian population was divided into 6 clusters, we calculated a log-ratio between AF in each cluster (RUS_cl) and gnomAD NFE AF for all 265,624 Finnish enriched variants. If a variant AF in the cluster was equal to zero, we considered the log-ratio equal to −5.

Additionally, we calculated the log-ratio between gnomAD East Asians AF and NFE AF. If variant's AF in the East Asians was equal to zero, we considered the log-ratio equal to −5. Subsequently, all variants with a log-ratio equal to −5 were considered absent in the East Asian population. We defined variants with a log-ratio between RUS AF and gnomAD NFE AF greater than 2 as Russian-enriched variants ($N = 44,963$).

### GWAS and other genetic data analyses
Original dataset of 823 phenotypes was subjected to a quality filtration prior to GWAS using R-libraries: "dplyr" (v1.0.0) and "tidyr" (v1.1.1)[42,43]. Only phenotypes with more than 200 individuals were considered for genetic analysis. The final number of phenotypes was 464. For several continuous phenotypes we filtered out the 5th and 95th quantiles of original phenotypic distribution to remove outliers suspected to be technical artifacts of lab value measurements (Supplementary Data 5). For all continuous phenotypes we calculated inverse rank-normal transformed (IRNT) versions. IRNT versions were not trimmed. For 28 phenotypes which characterized 6-year changes only IRNT versions were presented due to a small number of samples. The final GWAS data set included 3880 unrelated individuals and 7,565,503 variants (MAF > 0.01, HWE > 0.0001). As some phenotypes were collected only for a part of the initial cohort, additional quality control was performed (MAF > 0.01, HWE > 0.0001) for each phenotype independently. Linear and logistic regression models were used for continuous/categorical and binary phenotypes, respectively. All models were adjusted for sex, age, and PC1-PC4. Some phenotypes were also adjusted for history of medical treatment (Supplementary Note 15).

The GWAS results were annotated with VEP[44] and integrated into a PheWeb database[45]. We provide access through an online portal, Biobank Russia: https://biobank.almazovcentre.ru. Furthermore, genetic correlations were calculated between all pairs of phenotypes using LD-score regression, and significant values are shown in the PheWeb database[46,47]. Additionally, LD-score regression was used to calculate heritability of phenotypes from Biobank Russia and genetic correlation between corresponding phenotypes from UK Biobank and FinnGen.

We used POSTGAP[48] and GPrior[49] for gene mapping and prioritization in GWAS to confirm our findings in a case study of smoking status phenotypes. The R "ieugwasr" (v0.1.5) was used to retrieve FinnGen PheWas using the batch "finn-b"[50] for replication. R "TwoSampleMR" (v0.5.6) was used to clump genome-wide significant Finnish enriched variants[51]. The map on Fig. 1a and Supplementary Fig. 14 was created using the basemap toolkit from Python3.6 "matplotlib" (v3.5.2) library[52].

### Reporting summary
Further information on research design is available in the Nature Portfolio Reporting Summary linked to this article.

## Data availability
Open sharing of individual-level clinical and genetic data or deposition of this data into publicly accessible databases is not permitted under the laws of the Russian Federation, therefore, providing access to the raw data is beyond the control of authors. All results of GWAS, PheWAS, allele frequencies, and aggregated data, along with visualization, are available on the Biobank Russia portal: https://biobank.almazovcentre.ru. To download the summary statistics, please select the phenotype of interest and the link will be provided on the specific phenotype page along with Manhattan plot and other GWAS details. Applications for raw genotype/phenotype data access will be subject to material transfer agreements and should be sent to the corresponding author: Mykyta Artomov Mykyta.artomov@nationwidechildrens.org.

## Code availability
All custom code used in this study is available at github: https://github.com/ArtomovLab/RUS_BB/.

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

## Acknowledgements

Genetic data generation and contributions of K.D., P.L., A.P., M.J.D. were supported by the Finnish Academy, Center of Excellence in Disease Genetics (to A.P.). D.U., N.K., O.R., A.L., M.B., E.M., E.K., A.E., K.T., V.R., O.M., O.F., An.K., Al.K., E.S. were supported by the Ministry of Science and Higher Education of the Russian Federation (Agreement # 075-15-2022-301 to E.S.). M.A. was supported by the Aging Biology Foundation. The authors thank Dr. Maxim Artyomov (WUSTL) for his support of the project and helpful discussions.

## Author contributions

M.A., M.J.D., A.P., Al.K., An.K., O.R., E.S. designed the study. M.A. supervised the study. O.R., M.B., E.M., E.K., A.E., K.T., S.K., N.P., A.A., E.Bar., E.Baz.,O.B., E.V., A.B., N.C., A.D., E.L., I.B., R.L., D.D. recruited patients and collected biospecimens. K.D., P.L., O.M., O.F., An.K. managed the biobank and biospecimen. D.U., O.R., M.B., R.S., A.S., E.M., E.K., A.E., K.T. managed and curated the phenotypic data. D.U., N.K., A.L., V.R., I.M., M.A. analyzed the data. D.U., N.K., M.A. designed and created the PheWeb resource. D.U., O.R., M.J.D., M.A. wrote the manuscript. A.P., M.J.D., An.K., Al.K., E.S., M.A. acquired funding. All authors reviewed and approved the manuscript.

## Competing interests

M.J.D. is a founder of Maze Therapeutics. The remaining authors declare no competing interests.

## Additional information

**Supplementary information** The online version contains
supplementary material available at

Mykyta Artomov.

**Peer review information** *Nature Communications* thanks the anon-
ymous reviewers for their contribution to the peer review of this work. A
peer review file is available.

