## [Peer Review File · Nature Communications]

Complex Trait Susceptibilities and Population Diversity in a Sample of 4,145 RussiansEditorial Note: Parts of this Peer Review File have been redacted as indicated to remove third-party material where no permission to publish could be obtained.

Reviewer #1 (Remarks to the Author):

Usoltsev et al. reported a GWAS study in Russian populations. Given the increasing demand for genomic data and its analysis from diverse populations, and the relatively few reports analyzing Russian individuals to date, this study is likely to be welcomed by the scientific community. However, I have several concerns primarily about the quality control (QC) of the data sets.

It appears that imputation was carried out immediately after minimal QC, with additional QC processes applied post-imputation. However, standard practice generally involves conducting comprehensive QC on genotypes prior to performing imputation.

Furthermore, population genetics analysis typically employs experimentally genotyped data rather than imputed data. This may affect PCA, ADMIXTURE, IBD sharing, and Treemix analyses. I strongly suggest reperforming these analyses using genotyped high-confidence variants.

The initial QC identified a discrepancy between reported and genetic sex in 252 samples (5.3%)—a figure that seems excessively high given that the UK Biobank reported a discrepancy rate of around 0.1%. This could suggest possible sample mix-up or inaccurate inference of genetic sex due to noisy intensities in X/Y chromosomal SNPs. While the latter issue can be addressed with careful data analysis, the former may not be as easily resolved. I urge the authors to detail the measures taken to ensure minimal risk of sample mix-up.

The PCA should be reperformed with high-confidence genotyped variants. Previous studies have demonstrated that Russians in Moscow form a distinct cluster from UK or Italian populations (Heath et al. *EJHG* 2008; 16: 1413-29), and that several Russian populations are distant from the Finnish population (Khrunin et al. *PLoS One* 2013; 8: e58552). The authors' results do not align with these prior findings, which may be attributed to their procedure to create data sets for this analysis. This may also impact the ADMIXTURE and IBD-sharing analyses.

With regard to allele frequency analysis, I think such analysis requires precise information on allele frequency; imputed data are unlikely to provide this level of detail, even if a very high R² threshold is chosen. This analysis would ideally be conducted using whole genome sequence data.

Finally, in the GWAS, please check lambda_GC or LDSC for QC. I am somewhat concerned that the PCA plot (Fig 2A) suggests the existence of population structure, yet no effort appears to have been made to assess the impact of this issue. It would also be prudent to check the genetic correlation for the same/similar traits with FinnGen or UKB summary statistics.

Reviewer #2 (Remarks to the Author):

This is a well-done albeit preliminary and descriptive study, admittedly consistent with the author's discussion of the initial views into the dataset. Notably, Russian populations remain understudied in human genetics. This project provides an opportunity to describe the potential for human genetics studies using modern data platforms and data techniques.

Overall, the paper provides insights into a new dataset but really is limited on scientific take-homes to other researchers. Overall the existence of the dataset would be of high value given the under-represented nature of the population but discussion and conclusions are overall relatively short.

We confirmed the biobank portal is active and accessible to external researchers.

Major points:

How could differences in ascertainment bias affect population structure? Particularly the finngen microarray or other population-specific strategies, with HRC imputation. Could this affect results? If not it is worth mentioning it.

Treemix analyses would benefit from including additional branches and discussing the opportunities there given the discussed admixture process in the various Russian clusters.

Minor points:

“Ethnical” in the title should be reworded

Line 247: local ancestry in this context could confuse readers given other definitions. Suggest “geographically proximate ancestries” if that sounds reasonable to the authors.

Line 352: please give specifics of the summary statistics for the novel findings even if they still require replication.

We would like to thank the Editors and Reviewers for their valuable feedback and thorough evaluation of our manuscript. We have undertaken substantial revisions to improve the manuscript following the guidelines provided in the Reviewers' comments.

We hope that with all additional experiments and updated explanations in the manuscript you will find the manuscript to be significantly improved.

REVIEWER COMMENTS

Reviewer #1 (Remarks to the Author):

Usoltsev et al. reported a GWAS study in Russian populations. Given the increasing demand for genomic data and its analysis from diverse populations, and the relatively few reports analyzing Russian individuals to date, this study is likely to be welcomed by the scientific community. However, I have several concerns primarily about the quality control (QC) of the data sets.

1. It appears that imputation was carried out immediately after minimal QC, with additional QC processes applied post-imputation. However, standard practice generally involves conducting comprehensive QC on genotypes prior to performing imputation.

We apologize for the confusion. We have updated the main and supplementary text to include a comprehensive information on QC steps conducted prior to imputation that were performed in the original submission, but omitted from the text (see **Main text, Genetic data generation and imputation; Sup. Materials, Genotype imputation**).

Furthermore, following the recommendations from the reviewer we have conducted an extensive evaluation of imputation quality and compared the imputation pipeline that was used originally with the one suggested by the reviewer. Briefly, we have not found any statistical differences between the approaches and confirmed that none of the associations in the GWAS part of the manuscript was affected by imputation and pre-imputation QC pipeline changes.

Moreover, we confirmed good quality of imputation by masking the directly genotyped variants and then imputing them. The relevant results were added to the Supplementary materials (**Sup. Figure 1**). Below we provide a summary of the additional experiments.

Additional experiments

We agree with the Reviewer that some of the post-imputation QC steps could be done prior to imputation, in particular, additional sample filtering. However, we emphasize that it would not have any significant impact on imputation results. To prove our point we conducted several additional experiments.

We took 190 duplicated individuals discovered after kinship analysis and 252 samples with mismatch between detected and reported sex, and excluded them from the original dataset. Further, such filtered dataset were processed using the same QC steps and underwent pre-phasing and imputation with the same random seed number (to eliminate difference explained by the statistical noise). After imputation, the original dataset was compared to the one lacking duplicated and sex mismatch samples and the impact of removed samples on imputation results was evaluated.

Firstly, we compared DR2 values between these two datasets (See **Review figure 1a**). The DR2 values exhibited an extremely close resemblance ($R^2=0.95$, $p\text{-value} < 2 \times 10^{-16}$), with a concordance rate of 0.93 (DR2 difference < 0.1). Thus, estimated imputation quality metrics did not considerably differ for the vast majority of imputed variants.

Further, we extracted variants with $DR2 \geq 0.8$ for original dataset and $DR2 < 0.8$ for filtered dataset - *Set 1* ($n=296,684$, See the area 1 on **Review figure 1a**). We hypothesized that such variants could potentially possess a risk of false positive signal in GWAS, because they could potentially be filtered out if we would use a filtered dataset for further analysis. We checked whether such variants reached any statistical significance in any of GWAS presented in the manuscript. None of them reached genome-wide significance in any GWAS. Thus, presence of additional samples during imputation did not considerably influence association results.

Additionally, we took variants which were classified as well-imputed ($DR2 \geq 0.8$) in both datasets - *Set 2* ($n=10,156,657$, See the area 1 on **Review figure 1a**) - and calculated exact R^2 values between two vectors of imputed dosages for each variant [1]. All of the variants showed extreme concordance between two datasets (See **Review figure 1b**). Thus, such variants that were passed DR2 filter in both imputation results showed extreme concordance.

Finally, we assessed the difference in imputation quality by masking approach. We masked 5% ($n=23,704$) of directly genotyped variants (we sampled them chromosome-wise to account for non-uniform distribution of variants between chromosomes) to mimic the absence of these variants in both sets and further used them to assess imputation quality using IQS metric [1,3-5]. IQS estimates for masked variants were extremely resemblant between two datasets ($R^2=0.99$, $p\text{-value} < 2 \times 10^{-16}$). Also, we did not find any statistically significant difference in imputation quality between original and filtered datasets for the whole minor allele frequency spectrum (See **Review figure 1c**). Thus, using a masking approach, we comprehensively estimated imputation quality for both sets and did not find any significant difference.

In summary, our analysis indicates that the exclusion of samples filtered out after imputation did not yield any statistically significant impact on the ultimate imputation results.

Review figure 1. Experimental results. (a) - comparison of DR2 values between original and filtered dataset (we keep only imputed variants), density depicts number of variants in the region, Set 1 and 2 are explained in the text; (b) - Mean R² values and corresponding 95% CIs for variants with DR₂ ≥ 0.8 in both datasets; (c) - Comparison of mean imputation quality score (IQS) between original and filtered datasets for different minor allele frequency groups;

References:

- [1] - Kolosov N, Rezapova V, Rotar O, Loboda A, Freylikhman O, et al. (2022) Genotype imputation and polygenic score estimation in northwestern Russian population. PLOS ONE 17(6): e0269434. <https://doi.org/10.1371/journal.pone.0269434>
- [2] - Browning BL, Zhou Y, Browning SR. A One-Penny Imputed Genome from Next-Generation Reference Panels. Am J Hum Genet. 2018;103: 338–348. pmid:30100085
- [3] - Rowan TN, Hoff JL, Crum TE, Taylor JF, Schnabel RD, Decker JE. A multi-breed reference panel and additional rare variants maximize imputation accuracy in cattle. Genet Sel Evol. 2019;51: 77. pmid:31878893
- [4] - Ramnarine S, Zhang J, Chen L-S, Culverhouse R, Duan W, Hancock DB, et al. When Does Choice of Accuracy Measure Alter Imputation Accuracy Assessments? PLoS One. 2015;10: e0137601. pmid:26458263
- [5] - Lin P, Hartz SM, Zhang Z, Saccone SF, Wang J, Tischfield JA, et al. A new statistic to evaluate imputation reliability. PLoS One. 2010;5: e9697. pmid:20300623

2. Furthermore, population genetics analysis typically employs experimentally genotyped data rather than imputed data. This may affect PCA, ADMIXTURE, IBD sharing, and Treemix analyses. I strongly suggest reperforming these analyses using genotyped high-confidence variants.

Following Reviewer's suggestion, we have found several approaches for the data pre-processing prior to the population genetics analyses. For example, the study by Elghzaly et al (PMID: 36324510) suggests a very similar approach to the described by Reviewer and the study by Martin et al (PMID: 29706349) uses the post-imputation dataset in the analysis, consistently with our approach.

With multiple analyses pipelines described in the literature, we conducted extensive additional work to show that in our dataset the population genetics analyses using only directly genotyped variants yield the same results as described in the original

manuscript. **Review Figure 2** below shows the analyses presented in the Figures 2 and maximum likelihood tree from the Figures 3 of the original manuscript, but performed using only directly genotyped variants.

We also provide a comparison between genetic ancestry clustering obtained using all variants and only genotyped variants (**Review Figure 2E**).

Briefly, the results have not changed in any of the analyses and all of the qualitative outcomes remain without changes. The analyses conducted with imputed dataset provide better resolution due to a greater number of variants available for analysis, therefore, we kept the original figures with population genetics analyses, but added the Review Figure 2 to the supplement as evidence of the robustness of our results.

Review Figure 2. Populational analysis using only genotyped variants. (A) Principal component analysis with labeling indicating sample collection region; **(B)** Joint principal component analysis with 1000 Genomes cohort; **(C)** Joint principal component analysis with European subsample of 1000 Genomes cohort; **(D)** Nonoverlapping clustering of the Russian cohort in the PCA space; **(E)** Correlation matrix between clusters identifying using

all variants and only high quality genotyped variants; (F) Admixture analysis for each of six populational clusters in the Russian cohort, samples are arranged with respect to their PC1 coordinate; (G) Hierarchical clustering of the Russian cohort with 1000 Genomes subpopulations with respect to the sharing of IBD regions; (H) Joint principal component analysis with Russians and several neighboring populations from EGDP; (I) TreeMix analysis of the relatedness between 1000 Genomes subpopulations, EGDP populations, HGDP Russians and clusters within the Russian cohort.

3. The initial QC identified a discrepancy between reported and genetic sex in 252 samples (5.3%)—a figure that seems excessively high given that the UK Biobank reported a discrepancy rate of around 0.1%. This could suggest possible sample mix-up or inaccurate inference of genetic sex due to noisy intensities in X/Y chromosomal SNPs. While the latter issue can be addressed with careful data analysis, the former may not be as easily resolved. I urge the authors to detail the measures taken to ensure minimal risk of sample mix-up.

Thank you for pointing out the potential issue of sex-mismatch and sample mix-up. We have taken several measures to demonstrate the absence of any systematic errors in our dataset.

Initially, we observed 252 cases of Sex-mismatch. 102 of these samples were not duplicated. 152 samples had duplicates and using the genetic relatedness analysis we identified 67 independent samples (some samples had more than one duplicate). Therefore the real mismatch rate is smaller than we reported originally and equal to $102+67 = 169$ which is a 3.6%. All of the duplicates were removed from the analysis in the original manuscript, so this observation only relates to the way we calculated the mismatch rate.

Second, we implemented a control procedure to ensure there were no systematic errors in our sample mapping. We created a pool of 44 pairs of putatively related individuals identified through the clinical records. We specifically looked for putative father-child pairs as they could be approximated by the matching last names and child's patronymic (first name of the father).

Out of 44 putative pairs, we successfully identified 31 first-degree relatives, 1 second-degree relatives. Remaining 12 pairs were followed by the clinical team of which 10 have confirmed the absence of the relatedness (random match between the names) and 2 pairs did not respond to the follow-up request. In total we confirmed 32 out of 32 suspected relatives and confirmed the absence of relatedness among 10 pairs that were in fact not related. Which makes the possibility of systematic sample mismatch neglectably small, especially since analyzed sample pairs were distributed across the 96-well plates used for genotyping.

Next, we analyzed the pattern of sex mismatch observations in the 96-well plates that entered the genotyping pipeline. We identified 12 plates from a total of 43 that had more than 5 sex-mismatches each. We marked 937 samples on these 12 plates as potentially compromised due to the presence of sex-mismatched samples on the plates. We conducted a comparison of allele frequencies between these 937 samples and samples from the remaining 31 plates (**Review Figure 3**). This analysis revealed that no systematic error was present among the samples.

Finally, we confirmed that new associations reported in our manuscript could not be attributed to the 96-well plate origin of the sample. We did so by comparing the allele frequencies of GWAS hits that were highlighted in our manuscript (Supplementary Figure S17). This analysis employed a logistic model that incorporated variables such as compromised or non-compromised PLATE (0,1), number of alleles, phenotype, age, sex, and principal components (PC1 to PC4). Our analysis did not detect any statistically significant differences between allele frequencies for specific traits: Uric acid (rs4697701, $p=0.296$), LDL (rs7412, $p=0.28$), Abdominal obesity (rs56046524, $p=0.537$), Smoking initiation (rs13266066, $p=0.65$), Smoking status (rs7961991, $p=0.389$), and High blood pressure in the second half of pregnancy (rs11948871, $p=0.054$).

We added the description of this work to the supplement and believe that such an extensive investigation and evidence is sufficient to confirm the robustness of our findings and eliminate the possibility of the false-positive discoveries in GWAS.

Review Figure 3. Comparison between the allele frequencies of samples obtained from plates with over 5 sex-mismatches and those from plates with fewer than 5 sex-mismatches

4. The PCA should be reperformed with high-confidence genotyped variants. Previous studies have demonstrated that Russians in Moscow form a distinct cluster from UK or Italian populations (Heath et al. *EJHG* 2008; 16: 1413-29), and that several Russian populations are distant from the Finnish population (Khrunin et al. *PLoS One* 2013; 8: e58552). The authors' results do not align with these prior findings, which may be attributed to their procedure to create data sets for this analysis. This may also impact the ADMIXTURE and IBD-sharing analyses.

Thank you for addressing the problem of the genetic distance between the European, Finnish and Russian populations in your comment. As shown in response to

points 1-2, we did not find a fundamental difference between the use of only genotyped variants and all variants, therefore the difference pointed by Reviewer could not be attributed to technical aspects of our work.

We would like to highlight that in studies suggested by Reviewer, the analyzed Russian-descent individuals were selected from a very homogeneous population group. This can be confirmed by the fact that the variance within the Russian cluster on the PCA in Heath et al is extremely small compared to the known ethnic diversity of the population (PMID: 30902755). Our study included a random sample of the 3 metro areas and we expectedly see significant variance even within our sample.

Furthermore, we observe no overlaps of our dataset at PC1-PC2 with the CEU, GBR, IBS, and TSI populations (**Figure 2A-C**). This result aligns with the findings presented by Heath et al. Moreover, our IBD analysis highlights the emergence of distinct and independent clusters, namely cl1:cl5 and the Finnish population, which stand apart from the European population. Of note, the Finnish population was not a part of the analysis in Heath et al.

Second, the Russians included in the HGDP dataset exhibit a complete alignment with our 4th cluster when their positions are examined across PC1:PC2 (**Sup. Fig. S21**). Additionally, they show a concurrence with cluster 4 in the Treemix analysis. It is noteworthy that the Russians from the HGDP were recruited from the north-east region from Saint-Petersburg. This 4th cluster showcases a significant admixture of the Finnish population and has a larger presence in Saint-Petersburg than in other regions

Third, our recruitment strategy focused on three major cities situated in close proximity to regions where Ugro-Finnic minorities resided (**Review Figure 4**). This approach enabled us to capture not only Ugro-Finnic minorities but also encompassed a broad representation of ethnicities present within these urban centers. Khrunin et al. predominantly recruited individuals from small towns situated in different regions: central east and north east Russia. Since Russia is a multinational country, the point of recruitment of people will strongly affect the final ethnic composition (PMID: 30902755).

Fourth, in our principal component analysis involving both Russians and a European subset from the 1000 Genomes dataset, we observed an intersection between the Finnish population and cluster 4 of the Russian population in the PC1:PC2 plot. Notably, at PC3:PC4, a pronounced divergence between the Russian and Finnish populations became evident (**Review Figure 5**). This divergence underscores our ability to differentiate between the Finnish and Russian populations on higher principal components.

Thus, there is a notable overlap between previous studies and our results, yet, the differences could be adequately explained by the approach to cohort recruitment.

[Redacted]

Review Figure 4. The map of Finno-Ugric languages [Redacted]

Review Figure 5. PCA of joint Russian individuals with Europeans from 1000 Genomes

5. With regard to allele frequency analysis, I think such analysis requires precise information on allele frequency; imputed data are unlikely to provide this level of detail, even if a very high R² threshold is chosen. This analysis would ideally be conducted using whole genome sequence data.

Thank you for raising your concerns regarding the quality of imputed data. We share your perspective that utilizing whole genome sequencing (WGS) data for allele frequency analysis would yield more accurate outcomes. Nonetheless, it's important to

highlight that the imputed allele frequencies we employ in subsequent analyses maintain a high level of quality. As evidence of this, we conducted a masking experiment to thoroughly assess imputation quality and allele frequency consistency (see **Review Figure 6**). Further, we conducted additional allele frequency analysis using only directly genotyped variants, thus eliminating impact of potentially biased imputation (see **Review Figure 7**). We have added corresponding sections to the supplementary text (see **Sup. Figure 1; Sup. Materials, Genotype imputation**).

In short, imputation quality score (IQS) were more than 0.8 for the majority of masking variants (74%), slight decrease in median IQS value were observed only for less frequent variants (MAF < 0.05), however, still exceeding considerable high values (IQS(median)=0.86) (see **Review Figure 6a**). Additionally, only 720 (3%) out of 23,704 were classified as discordant in terms of imputed allele frequency ($\log_{FC} > 5$ or $\log_{FC} < -5$ or AF difference > 0.10) (see **Review Figure 6b**).

Frequency analysis on directly genotyped variants replicated the enrichment for all variants (**Review Figure 7A**) as it was shown previously in the original manuscript (**Fig 3A**). For variants that originated specifically in the Finnish population we observed a noticeable enrichment in cluster 4 (**Review Figure 7B**) which is aligned with the original work (**Fig. 3B**).

Review figure 6. Experimental results. (a) - Imputation Quality Score derived for masked variants for several minor allele frequency groups; (b) - Allele frequency comparison between masked and imputed variants; MAF - minor allele frequency, AF - alternative allele frequency.

FINNISH ENRICHED VARIANTS*
ONLY GENOTYPED VARIANTS

Review figure 7. (A) Distribution of enrichment of the genotyped Finnish-enriched variants across clusters in the Russian cohort, Finnish and East Asian cohorts from gnomAD; **(B)** Distribution of enrichment of the Finnish-enriched variants that are not found in East Asian population in gnomAD across clusters in the Russian cohort, Finnish and East Asian cohorts from gnomAD.

6. Finally, in the GWAS, please check λ_{GC} or LDSC for QC. I am somewhat concerned that the PCA plot (Fig 2A) suggests the existence of population structure, yet no effort appears to have been made to assess the impact of this issue. It would also be prudent to check the genetic correlation for the same/similar traits with FinnGen or UKB summary statistics.

Thank you for your concerns about the quality of our GWAS.

Initially, we examined lambdas for all of our GWAS. The median value of the lambda distribution was 1.004 which is in a good agreement with expected value of 1. Among the 518 traits analyzed, only one exhibited a lambda value exceeding 1.05 due to a very low case count (N=9), (**Review Figure 8**). This outcome validates the effectiveness of our variant quality control and confirms the absence of systematic bias in our GWAS.

Review Figure 8. Distribution of genomic inflation factors (lambda) across all performed GWAS.

Furthermore, following Reviewer's suggestion, we conducted an investigation of the genetic correlations between phenotypes from the Russian Biobank and their corresponding counterparts in the UK Biobank and FinnGen datasets. Initially, we excluded all GWAS that had non-reliable heritability estimates (confidence interval outside the range of [0,1]) and a heritability standard error more than 50%. This left us with a total of 40 GWAS. 31 of which were matched with corresponding traits from the UK Biobank (**Sup. Tab. S4**). For complex phenotypes, we compared them with all relevant components from the UK Biobank dataset. Altogether, we performed 39 pairwise comparisons. Out of 39 comparisons 31 were nominally significant ($p < 0.05$) and 26 passed the Bonferroni significance threshold ($0.05/39=0.00128$). From the FinnGen traits, we specifically chose dyslipidemia, hypertension, type 2 diabetes (T2D), obesity, myocardial infarction, ischaemic heart disease, anxiety, depression, smoking and sleep apnoea. We proceeded to examine the genetic correlation between these selected traits and their corresponding counterparts from our pool of 40 traits. We constructed 17 pairs and detected nominal significance for 14 of them while 10 passed the Bonferroni significance threshold ($0.05/17=0.00294$) (**Sup. Tab. S4**).

Reviewer #2 (Remarks to the Author):

This is a well-done albeit preliminary and descriptive study, admittedly consistent with the author's discussion of the initial views into the dataset. Notably, Russian populations remain understudied in human genetics. This project provides an opportunity to describe the potential for human genetics studies using modern data platforms and data techniques.

Overall, the paper provides insights into a new dataset but really is limited on scientific take-homes to other researchers. Overall the existence of the dataset would be of high value given the under-represented nature of the population but discussion and conclusions are overall relatively short.

We confirmed the biobank portal is active and accessible to external researchers.

Major points:

How could differences in ascertainment bias affect population structure? Particularly the finngen microarray or other population-specific strategies, with HRC imputation. Could this affect results? If not it is worth mentioning it.

Thank you for addressing the potential systematic errors that might arise from our genotyping microarray. We selected chip-independent HapMap variants and conducted an ADMIXTURE analysis (**Review Figure 9**). Our findings reveal that there is no noticeable reduction in Finnish haplotypes within clusters of the Russian population. This observation suggests that the genotyping microarray is unlikely to be the source of systematic errors. We have included the pertinent conclusions in the main text and provided the corresponding figure in the Supplementary materials.

Review Figure 9. Admixture analysis using only HapMap variants

Treemix analyses would benefit from including additional branches and discussing the opportunities there given the discussed admixture process in the various Russian clusters.

Thank you for your comment.

We incorporated all HGDP populations into our Treemix analysis (**Review Figure 10**). However, it's important to note that based on the geographical locations of these populations, they are all quite distant from Russia. This observation was further supported by our newly constructed maximum likelihood tree.

Review Figure 10. Treemix analysis with additional populations from HGDP

Minor points:

“Ethnical” in the title should be reworded

Thank you, the title was corrected.

Line 247: local ancestry in this context could confuse readers given other definitions. Suggest “geographically proximate ancestries” if that sounds reasonable to the authors.

Thank you, the sentence was re-written.

Line 352: please give specifics of the summary statistics for the novel findings even if they still require replication.

Thank you. We have included information about the leading variants in these GWAS in the main text.

Reviewer #1 (Remarks to the Author):

Usoltsev et al. made great efforts to validate the quality of their results. Martin et al (PMID: 29706349) imputed the Finngen data set using 1000 Genomes Project, which includes Finnish samples. Russians are an under-represented population in imputation reference panels. I had concerns about haplotype matching (similar to other studies of under-represented populations), but the authors demonstrated that the differences between the analysis results using genotyped and imputed data were negligible. Therefore, I can accept the results. Also, adding genetic correlation results is great.

Reviewer #3 (Remarks to the Author):

Usoltsev et al. present the first results of analyses of clinical and genetic data from three cities in western Russia that participated in Biobank Russia. As the other reviewers noted, the sample size of the GWAS is not enough to stand alone but will be useful in meta-analysis. I concur that the data will be welcomed as a valuable resource by the scientific community.

1. In my opinion, concerns regarding the use of imputed data are overstated. At the beginning of imputation ~15 years ago, both population geneticists and genetic epidemiologists were concerned about using imputed data. Over the last 10+ years, the use of imputed data in population genetics analysis by genetic epidemiologists has become routine and expected. Provided that the genotype data that form the scaffold for imputation have good quality and that the imputation performed reasonably well, imputed data typically outperform genotype data in population genetics analyses. A key reason is that genotype data suffer from ascertainment bias resulting from the selection of variants to be interrogated by the array, whereas imputed data approximate whole genome sequences. The analyses with genotyped data that Reviewer #1 requested that involve repeating analyses originally performed with imputed data are not guaranteed to work as the reviewer intended. In response to Reviewer #2, the use of imputed data is recommended over genotype data with respect to ascertainment bias, so long as the HRC contains relevant haplotypes.

2. Concerns about PCA, population structure, and confounding in the GWAS are misplaced. PCs 1 and 2 show population-level structure. In Supplementary Figure S5, PCs 3 and 4 show patterns consistent with family-level structure, including known and cryptic relatedness. The GWAS already included PCs 1-4 (line 237). The inclusion of λ_{GC} is good practice, and the authors have responded positively. The PCA plots should also include the percent of variance explained by PC. The important issue that is not addressed is admixture, which is evident in Figure 2E and which the inclusion of PCs in the GWAS accounts for in terms of genome-wide ancestry but not in terms of locus-specific ancestry.

3. I do not recommend taking a quantitative trait, discretizing it into deciles (lines 230-231), and then trimming the bottom and top 5th percentiles (lines 231-232). The resulting distribution has been truncated. I would not discretize a continuous variable in the first place. After trimming, it is unclear if the authors analyzed quantitative traits as continuous variables (using the original values) or if the authors analyzed quantitative traits as categorical variables.

4. With respect to Reviewer #2's comments regarding adding additional branches and discussing admixture in the TreeMix analysis, the authors responded by adding HGDP populations. I think readers will benefit from a discussion of cluster 6. The admixture bar plots look like what have been described in the literature based on samples from Kazakhstan, Kyrgyzstan, Tajikistan, Turkmenistan, and Uzbekistan (for example, DOIs 10.1038/nature09103, 10.1093/molbev/msr221, 10.1371/journal.pgen.1003634, 10.1371/journal.pone.0076748, 10.1038/nature12736, and 10.1038/s41598-017-01837-7). This interpretation is reinforced by the analyses presented in Figure 3D and Supplementary Figures S21 and S22, with the placement of samples from cluster 6 near samples from Central Asia. The readers should feel assured that the evidence for structure is genuine

(and unsurprising given the geographic proximity of the Samara and Orenburg sampling sites to Kazakhstan) and reflects some of the genetic diversity present in Russia outside of ethnic Russians.

5. Like Reviewer #1, I was puzzled by the seemingly high number of sex mismatches. The authors responded that the number was inflated because of sample duplication. It is not clear what the original problem was. I am unfamiliar with the FinnGen array, but it is not immediately apparent to me that there is a technical issue. I suspect bad samples, in which case the only thing to do is to discard them and move on.

6. The descriptor CEU does not stand for "Central Europeans" (lines 268, 270, 279, and 638 in the main text and pp. 17 and 18 in the supplement). CEU stands for "Utah residents (CEPH) with Northern and Western European ancestry".

We would like to thank the Editors and Reviewers for their valuable feedback and thorough evaluation of our manuscript.

REVIEWER COMMENTS

Reviewer #1 (Remarks to the Author):

Usoltsev et al. made great efforts to validate the quality of their results. Martin et al (PMID: 29706349) imputed the Finngen data set using 1000 Genomes Project, which includes Finnish samples. Russians are an under-represented population in imputation reference panels. I had concerns about haplotype matching (similar to other studies of under-represented populations), but the authors demonstrated that the differences between the analysis results using genotyped and imputed data were negligible. Therefore, I can accept the results. Also, adding genetic correlation results is great.

Thank you!

Reviewer #3 (Remarks to the Author):

Usoltsev et al. present the first results of analyses of clinical and genetic data from three cities in western Russia that participated in Biobank Russia. As the other reviewers noted, the sample size of the GWAS is not enough to stand alone but will be useful in meta-analysis. I concur that the data will be welcomed as a valuable resource by the scientific community.

We greatly appreciate the detailed review of our work.

1. In my opinion, concerns regarding the use of imputed data are overstated. At the beginning of imputation ~15 years ago, both population geneticists and genetic epidemiologists were concerned about using imputed data. Over the last 10+ years, the use of imputed data in population genetics analysis by genetic epidemiologists has become routine and expected. Provided that the genotype data that form the scaffold for imputation have good quality and that the imputation performed reasonably well, imputed data typically outperform genotype data in population genetics analyses. A key reason is that genotype data suffer from ascertainment bias resulting from the selection of variants to be interrogated by the array, whereas imputed data approximate whole genome sequences. The analyses with genotyped data that Reviewer #1 requested that involve repeating analyses originally performed with imputed data are not guaranteed to work as the reviewer intended. In response to Reviewer #2, the use of imputed data is recommended over genotype data with respect to ascertainment bias, so long as the HRC contains relevant haplotypes.

In our earlier work (PMID: 35763490), we have assessed the quality of genotype imputation in a small cohort (~300 subjects) of Russian origin. We have done so by “masking” the directly genotyped variants and then imputing them using different

reference panels. HRC significantly outperformed 1000 Genomes in the quality of imputation and the number of well-imputed variants. Therefore, we do not anticipate creating any bias by imputing genotypes in our dataset with HRC as a reference panel. Subsequently, as pointed out by the Reviewer, the results of population genetics analyses do not differ between imputed and non-imputed datasets.

2. Concerns about PCA, population structure, and confounding in the GWAS are misplaced. PCs 1 and 2 show population-level structure. In Supplementary Figure S5, PCs 3 and 4 show patterns consistent with family-level structure, including known and cryptic relatedness. The GWAS already included PCs 1-4 (line 237). The inclusion of λ_{GC} is good practice, and the authors have responded positively. The PCA plots should also include the percent of variance explained by PC. The important issue that is not addressed is admixture, which is evident in Figure 2E and which the inclusion of PCs in the GWAS accounts for in terms of genome-wide ancestry but not in terms of locus-specific ancestry.

Thank you! We have estimated the fraction of variance explained by PCs and added this to the figures.

We agree with the Reviewer that inclusion of admixed individuals into the analysis may impose potential risk of false positive associations. Therefore, for novel associations that we report in the manuscript, we have checked the consistency of effects across all identified genetic ancestry clusters identified in our data. Sup. Figure S26 shows that effects are consistent across all subpopulations with some of the clusters having too few study subjects to show the significant deviation of the effect size from 0. We also added a sentence to the discussion section to acknowledge the potential limitation of the study associated with a small number of admixed individuals included in GWAS.

3. I do not recommend taking a quantitative trait, discretizing it into deciles (lines 230-231), and then trimming the bottom and top 5th percentiles (lines 231-232). The resulting distribution has been truncated. I would not discretize a continuous variable in the first place. After trimming, it is unclear if the authors analyzed quantitative traits as continuous variables (using the original values) or if the authors analyzed quantitative traits as categorical variables.

We apologize for the misunderstanding. First, quantitative traits were analyzed as quantitative without discretizing. Only for several continuous phenotypes listed in Sup. Tab. S2 we trimmed bottom and top 5th percentiles due to the presence of individuals with abnormally high or low lab-generated values which were assumed to be technical artifacts. This was clarified in the methods and at most may result in the lack of power for these specific phenotypes without the risk of false positive results.

Originally, we also discretized continuous phenotypes, to keep individuals with unusually high/low measurements in the analysis and ensure they would not cause false positive associations. However, now to address your suggestion, we have removed discretized analysis of continuous phenotypes and replaced it with an inverse rank normalization for

continuous traits, similarly to how it was done in the UK biobank analysis performed by Neale Lab (<http://www.nealelab.is/uk-biobank>). These results are now available through the Biobank Russia PheWeb portal as well. While we have not found significant associations, this should ensure the most accurate approach to continuous data analysis.

4. With respect to Reviewer #2's comments regarding adding additional branches and discussing admixture in the TreeMix analysis, the authors responded by adding HGDP populations. I think readers will benefit from a discussion of cluster 6. The admixture bar plots look like what have been described in the literature based on samples from Kazakhstan, Kyrgyzstan, Tajikistan, Turkmenistan, and Uzbekistan (for example, DOIs 10.1038/nature09103 (Jewish), 10.1093/molbev/msr221 (Caucasus), 10.1371/journal.pgen.1003634 (Mongolians), 10.1371/journal.pone.0076748 (Afghan), 10.1038/nature12736 (Siberian), and 10.1038/s41598-017-01837-7 (language)). This interpretation is reinforced by the analyses presented in Figure 3D and Supplementary Figures S21 and S22, with the placement of samples from cluster 6 near samples from Central Asia. The readers should feel assured that the evidence for structure is genuine (and unsurprising given the geographic proximity of the Samara and Orenburg sampling sites to Kazakhstan) and reflects some of the genetic diversity present in Russia outside of ethnic Russians.

Thank you! Indeed, cluster 6 is a combination of Asian and European haplotypes. This mix is particularly noticeable in the subgroup of cluster 6 with the highest PC1 values. This subgroup demonstrates a strong affinity with Central Asian populations. We have added extra statements to the Results section to clarify the origin of cluster 6. We also have expanded the discussion section to include the mentioned citations.

5. Like Reviewer #1, I was puzzled by the seemingly high number of sex mismatches. The authors responded that the number was inflated because of sample duplication. It is not clear what the original problem was. I am unfamiliar with the FinnGen array, but it is not immediately apparent to me that there is a technical issue. I suspect bad samples, in which case the only thing to do is to discard them and move on.

Thank you! We demonstrated that there was no systematic sample mislabeling and all the known family/phenotypic relationships were preserved during the genetic data generation process. Yet, samples with sex mismatch and all duplicates were excluded from the study, resulting in the higher-than-expected drop out rate, but ensuring only good quality data entered the analysis.

6. The descriptor CEU does not stand for "Central Europeans" (lines 268, 270, 279, and 638 in the main text and pp. 17 and 18 in the supplement). CEU stands for "Utah residents (CEPH) with Northern and Western European ancestry".

Thank you for pointing this out! The descriptions were updated.

Reviewer #3 (Remarks to the Author):

The authors' responses have exceeded my expectations. This revision satisfactorily address not only the comments from Reviewer #2 but also my comments.